# Molecular characterization of fowl adenovirus isolate of Malaysia attenuated in chicken embryo liver cells and its pathogenicity and immunogenicity in chickens

**Norfitriah Mohamed Sohaimi[1], Mohd Hair Bejo** [1,2]*****, **Abdul Rahman Omar[1,2], Aini Ideris[1,2], Nurulfiza Mat Isa[2,3]**

**1** Faculty of Veterinary Medicine, Universiti Putra Malaysia, Serdang, Selangor, Malaysia, **2** Institute of Bioscience, Universiti Putra Malaysia, Serdang, Selangor, Malaysia, **3** Faculty of Biotechnology and Biomolecular Sciences, Universiti Putra Malaysia, Serdang, Selangor, Malaysia

* mdhair@upm.edu.my

**Data Availability Statement:** All nucleotide sequences data are available from the GenBank

## Abstract

Fowl adenovirus (FAdV) is the causative agent of inclusion body hepatitis (IBH) in chickens with significant economic losses due to high mortality and poor production. It was objectives of the study to attenuate and determine the molecular characteristic of FAdV isolate (UPM1137) of Malaysia passages in primary chicken embryo liver (CEL) cells. The cytopathic effect (CPE) was recorded and the present of the virus was detected by polymerase chain reaction (PCR). Nucleotide and amino acid changes were determined and a phylogenetic tree was constructed. The pathogenicity and immunogenicity of the virus at passage 35 (CEL35) with virus titre of $10^{6.7}TCID_{50}$/mL was determined in day old specific pathogen free (SPF) chicks via oral or subcutaneous route of inoculation. The study demonstrated that the FAdV isolate was successfully propagated and attenuated in CEL cells up to 35th consecutive passages (CEL35) with delayed of CPE formation within 48 to 72 post inoculation (pi) from CEL20 onwards. The virus caused typical CPE with basophilic intranuclear inclusion bodies, refractile and clumping of cells. The virus is belong to serotype 8b with substitution of amino acid at position 44, 133 and 185 in L1 loop of hexon gene and in knob of fiber gene at position 348 and 360 at CEL35. It is non-pathogenic, but immunogenic in SPF chickens. It was concluded that the FAdV isolate was successfully attenuated in CEL cells with molecular changes in major capsid proteins which affect its infectivity in cell culture and SPF chickens.

## Introduction

Fowl adenovirus (FAdV) is an important causative agent of devastating disease known as inclusion body hepatitis (IBH) in poultry with significant economic impact due to high mortality and poor production [1, 2]. The virus is classified as *Aviadenovirus* genus under family of *Adenoviridae* with five molecular species group designated as letter A to E, and 12 serotypes [3]. Highly pathogenic FAdV is the primary pathogen of IBH and hydropericardium

database (accession numbers: KF866370, KY305950, KY305943, KY305944, KY305945, KY305946, KY305947, KY305948, KY305949, KY305951, KY305952, KY305953, KY305954, KY305955, KY305956, KY305957.

**Funding:** MHB received grant award with number 6369101 and 6364002 from Ministry of Education, Malaysia, and Ministry of Science and Technology, Malaysia; www.moe.gov.my (Ministry of Education), www.mestecc.gov.my (Ministry of Science and Technology). The funders had no role in study design, data collection and analysis, decision to publish, or preparation of the manuscript.

**Competing interests:** The authors have declared that no competing interests exist.

syndrome (HPS) due to FAdV-1, FAdV-8, FAdV-9 and FAdV-4 infection, respectively. The infection caused sudden onset of high mortality and immunosupresion in chickens [2, 4, 5, 6, 7]. In Malaysia, IBH outbreak was first reported by Hair-Bejo in 2005 [8] in commercial broiler chickens with 10% mortality. Since then, the number of IBH and gizzard erosion cases were increased due to FAdV Group E serotype 8b infection [9, 10]. Vaccination is not a common practice in the country and suitable vaccine against the disease is unavailable.

A few approaches of viral attenuation such as serial passages in cell culture or chicken embryos were reported previously for the development of live attenuated vaccine [11, 12]. It seems that avian origin cells such as primary chicken embryo liver (CEL) cells are the superior medium for FAdV adaptation as compared to mammalian cells [13, 14]. In addition, continuous cell line from chicken hepatoma cells and QT35 quail fibroblast cells, and mammalian cell line from Vero cells were also used previously [11, 15, 16].

The characterization of major capsid proteins are extensively used in determination of virulence gene in FAdV isolate [5, 17]. Mutation of nucleotide and amino acids at L1 loop region in hexon gene affect virus virulence and resulting attenuation of FAdV following passage in chicken embryonated eggs [10, 12]. However, the marker for adaptation and attenuation of FAdV in CEL cells remained unknown which typically important for localization of specific region encoded for virus infectivity. Previous mutation analysis basically performed solely on hexon gene without involvement of fiber gene which located at outermost part of virus capsid for primary attachment to host cells receptor [12]. Although FAdV routinely isolated in CEL cells, the impact of serial passages in this medium towards molecular changes in hexon and fiber gene specifically at nucleotide and amino acid level which may affect viral infectivity in the host are poorly known. It was objectives of the study to propagate and attenuate FAdV isolate (UPM1137) in primary CEL cells. The molecular characteristic of the virus at various passages was determined. The pathogenicity and immunogenicity of the virus isolate at passage 35 (CEL35) was determined in the specific pathogen free (SPF) chickens.

## Materials and methods

### Virus

FAdV isolate, UPM1137, was obtained from an outbreak of IBH and gizzard erosion in 27 weeks old commercial layer chickens with 2% mortality. Upon necropsy, the liver was pale and friable with erosion in koilin layer of gizzard. Both liver and gizzard samples were positive for FAdV by histological examination and conventional polymerase chain reaction (PCR) [18]. Liver from the infected chickens was collected and processed by three times frozen and thawed prior macerated with a sterile mortar and pestle for preparation 1 in 2 (w/v) suspension in sterile phosphate buffered saline (PBS, pH 7.4, 0.1M). For clarification, suspension was centrifuged at 381 x *g* for 30 minutes and the supernatant was collected and purified by filtration through 0.45μm syringe filter. Liver homogenates was treated with commercial antibiotic-antimycotic solution (GIBCO Laboratories, New York, NY, USA) at 1 in 10 (v/v) dilutions and incubated at 4°C for 1 hour prior inoculation [19]. The virus inoculum (0.1 mL) was then inoculated in SPF embryonated chicken eggs via chorioallantoic membrane (CAM) route. The eggs were candled daily for mortality. Severe hepatic necrosis was recorded in the dead embryos. The liver was harvested for the next passage (E2) in SPF eggs for the preparation of inoculum used in the present study. DOI: dx.doi.org/10.17504/protocols.io.7zjhp4n.

### Preparation of primary chicken embryo liver (CEL) cells

Primary CEL cell was obtained from liver embryos of 13 to 15 days old SPF embryonated chicken eggs. Liver was harvested aseptically using sterile forceps and washed twice with sterile

phosphate buffered saline (PBS, pH 7.4, 0.1M). The liver tissue was minced and trypsinized gently with 0.25% Trypsin-EDTA solution for 10 minutes. The suspension was passed through muslin cloth and centrifuged at 96 x $g$ for 5 minutes to obtain cell pellet. Trypsin was discarded and cell pellet was resuspended with fresh Dulbecco's Modification Eagle Medium (DMEM), enriched with 10% fetal bovine serum (FBS) and 1% penicillin-streptomycin antibiotic. Cell concentration was counted and adjusted to 5 x$10^6$ cells/mL. The cell suspension (5mL) was seeded into new 25cm$^2$ cell culture flasks and was kept under controlled atmosphere at 5% $CO_2$ incubator with 85%-90% humidity until confluent monolayer formed [14]. DOI: dx.doi. org/10.17504/protocols.io.7zchp2w.

## Propagation and attenuation of FAdV isolate in CEL cells

A confluent monolayer was washed twice with serum free medium and inoculated with 0.1mL of homogenate liver embryos and labeled as first passage (CEL1). Infected flasks were incubated at 37°C for 60 minutes for virus adsorption and added with maintenance medium containing 2% fetal bovine serum (FBS) under 37°C incubator. The cells were observed daily under inverted microscope for cytopathic effect (CPE) for 3 days post-inoculation (pi). Flasks with prominent CPE were harvested by 3 times repeated frozen and thawed prior centrifugation at 216 x $g$ for 10 minutes. Virus supernatant was collected and stored at -20°C prior inoculation for subsequence passages. Fresh confluent monolayer was prepared for each passage and inoculated with 0.1mL of viral supernatant and continued until 35$^{th}$ consecutive passage. For non-infected flasks, monolayer remained uninoculated and was used as control cells. DOI: dx.doi.org/10.17504/protocols.io.7zdhp26.

## Haematoxylin and eosin staining

For detection of intranuclear inclusion bodies, primary CEL cells were grown on sterile coverslips [20]. Briefly, the growth medium was aspirated out from confluent monolayer and washed twice with sterile medium free serum prior to inoculation of 0.1mL viral supernatant. Culture was added with maintenance medium, DMEM and 2% FBS and kept for 48 hours incubation, harvested and fixed in 10% buffered formalin for 5 minutes and stained with haematoxylin and eosin (HE). DOI: dx.doi.org/10.17504/protocols.io.7zghp3w.

## Virus titration

Viral supernatant from passage 5 (CEL5), CEL10, CEL15, CEL20, CEL25, CEL30 and CEL35 were determined for titration by 50% tissue culture infective dose (TCID$_{50}$) as described by Reed and Muench [21]. Supernatant was diluted for 10 folds serial dilution and inoculated onto fresh CEL cells in 96 wells plate and monitored daily for CPE. Virus titre was calculated and expressed as TCID$_{50}$/mL. DOI: dx.doi.org/10.17504/protocols.io.7zhhp36.

## DNA extraction and quantitation

DNA was extracted from 200μl of homogenate liver embryos (E2) and viral supernatant from propagated isolates in primary CEL cells from CEL1, CEL5, CEL10, CEL15, CEL20, CEL25, CEL30 and CEL35 using i-Genomic DNA Extraction Mini kit (iNtRON Biotechnology, Inc., Korea). The protocol was followed according to manufacture recommendations using spin column method. Extracted DNA was measured for DNA concentration by biophotometer (Eppendorf) at wavelength 260/280nm.

## PCR and cloning

Amplification of DNA was performed by two set of published hexon gene primers, H1/H2 and H3/H4 [22] and fiber gene primer, FibF/FibR, `FibF: 5`-GGTCTACCCCTTTTGGCTCC-3`and FibR: 5`-GCGTCGTAGATGAAGGGAGG-3`` [10] according to manufacture protocol (Bioline, UK). The PCR products were analyzed by electrophoresis in a 1% agarose gel stained with RedSafe™ Nuclei Acid Staining solution (iNtRON, Korea) at 70 volts for 45 minutes and visualized under U.V. transillumination. Purification of PCR products were carried out by using MEGAquick-spin™ Total Fragment DNA Purification kit (iNtRON) based on the manufacture recommendation. Purified PCR products from H1/H2 and FibF/FibR were cloned into the pCR™ 2.1-TOPO® vector using TOPO TA Cloning kit (Invitrogen, USA). Positive cloned was analyzed by colony-PCR prior plasmid extraction using DNA-spin™ Plasmid DNA Purification kit (iNtRON) and stored at -20˚C until used. DOI: dx.doi.org/10.17504/protocols.io.7zfhp3n.

## Sequencing and sequence analysis

Plasmid from original isolate and series of propagated isolates were subjected to DNA sequencing on an automatic sequencer, ABI PRISM 3730*xl* Genetic Analyzer (Applied Biosystems, USA) using the BigDye® Terminator v3.1 Cycle Sequencing Kit (Applied Biosystems, USA). Sequencing was conducted at least three times to obtain consensus sequences using universal primers, T7 promoter and M13R_pUC26 for both direction [10]. The sequence was verified as fowl adenovirus Group E based on NCBI BLAST GenBank. Nucleotide sequences were assembled, edited and analyzed using BioEdit Version 7.2.5. Sequence was translated into amino acid sequences using online ExPASy tool program. Multiple sequence alignment was carried out using ClustalW program in BioEdit version 7.2.5 package to compare nucleotide and deduced amino acids sequence between original isolate (E2) and propagated isolates in CEL cells. The E2 FAdV designated as UPM1137E2 with GenBank accession number KF866370 (hexon) and KY305950 (fiber). Sequence difference count matrix was carried out to calculate nucleotide and amino acid changes between each passage before and after attenuation. In addition, location for variable L1 loop hexon gene and knob region in fiber gene was identified by alignment with reference FAdV gene, HG strain [23]. L1 loop of hexon gene corresponding to residue 101 to 298 of HG strain, while for fiber gene as follows: Tail: 1–75, Shaft: 76–356, Knob: 369–523 [17]. DOI: dx.doi.org/10.17504/protocols.io.7zehp3e.

## Phylogenetic analysis

Twenty-eight published FAdVs hexon (Table 1) and fiber (Table 2) gene sequences were obtained from GenBank. Another two reference strains from different genus avian adenovirus was used which are Duck adenovirus (Atadenovirus) and Turkey adenovirus (Siadenovirus). Multiple sequence alignment of nucleotide and deduced amino acid were conducted using BioEdit Version 7.2.5 and Mega Version 5 software. A region of L1 loop in hexon gene with 198 amino acids and entire region of fiber gene was selected for construction of phylogenetic tree. Jones-Taylor-Thorton (JTT) model was used to compute distance matrix using MEGA software and followed by phylogenetic tree construction using Neighbour-joining method with 1000 boostrap replicates [9]. DOI: dx.doi.org/10.17504/protocols.io.7zihp4e.

## Pathogenicity and immunogenicity of the FAdV CEL35 isolate in SPF chickens

Twenty eight day-old SPF White Leghorn layer chickens were divided into three groups, namely groups A, B and C. Eight chickens each were assigned in groups A and B separately and twelve chickens in group C. All chickens in groups A and B were inoculated with 0.1mL

**Table 1. Thirty avian adenovirus strains for hexon gene nucleotide and amino acids were used in the phylogenetic tree analysis.**

| No. | Strains | Accession number | | Group[a] | References |
|---|---|---|---|---|---|
| | | Nucleotide | Amino acid | | |
| 1 | CELO | Z67970 | CAA91908 | A | S1 File [A] |
| 2 | PL/060/08 | GU952110 | ADF57268 | A | S1 File [B] |
| 3 | 340 | AF508952 | AAN77078 | B | S1 File [C] |
| 4 | TR22 | AF508953 | AAN77079 | B | S1 File [D] |
| 5 | KR5 | AF508951 | AAN77077 | C | S1 File [E] |
| 6 | B1-7 | KU342001 | ANG08820 | C | S1 File [F] |
| 7 | J2-A | AF339917 | AAL13220 | C | S1 File [G] |
| 8 | C2B | EU979377 | ACL68145 | C | S1 File [H] |
| 9 | SR48 | AF508946 | AAN77072 | D | S1 File [I] |
| 10 | P7-A | AF339915 | AAL13218 | D | S1 File [J] |
| 11 | SR49 | AF508948 | AAN77074 | D | S1 File [K] |
| 12 | **75** | AF508949 | AAN77075 | D | S1 File [L] |
| 13 | A-2A | NC_000899 | NP_050287 | D | S1 File [M] |
| 14 | 380 | KT862812 | ANJ02603 | D | S1 File [N] |
| 15 | 1047 | DQ323984 | ABD83863 | D | S1 File [O] |
| 16 | UF71 | EU979378 | ACL68146 | D | S1 File [P] |
| 17 | CR119 | AF508954 | AAN77080 | E | S1 File [Q] |
| 18 | YR36 | AF508955 | AAN77081 | E | S1 File [R] |
| 19 | B-3A | AF339922 | AAL13225 | E | S1 File [S] |
| 20 | 58 | AF508957 | AAN77083 | E | S1 File [T] |
| 21 | TR59 | EU979374 | ACL68142 | E | S1 File [U] |
| 22 | 764 | JN112373 | AER40292 | E | S1 File [V] |
| 23 | Australian FAdV vaccine | GU120268 | ADD49658 | E | S1 File [W] |
| 24 | 430–06 | GU120266 | ADD49656 | E | S1 File [X] |
| 25 | HG | GU734104 | ADE58399 | E | S1 File [Y] |
| 26 | UPM08158 | JF917238 | AEL21619 | E | S1 File [Z] |
| 27 | UPM08136 | JF917239 | AEL21620 | E | S1 File [AA] |
| 28 | UPM04217 | KU517714 | ANA50319 | E | S1 File [BB] |
| 29 | Duck adenovirus | KF286430 | AGS11274 | N/A | S1 File [CC] |
| 30 | Turkey adenovirus | AC_000016 | AP_000486 | N/A | S1 File [DD] |

N/A = not applicable

[a] FAdV comprised of 5 molecular groups designated as letter A, B, C, D and E.

FAdV isolate, UPM1137CEL35 with virus titre of $10^{6.7}TCID_{50}$/mL via oral and subcutaneous route, respectively at day old of age. Twelve chickens in group C remained uninoculated throughout the trial and acted as the control group. All chickens were monitored daily throughout 28 days post-inoculation (pi). Feed and water were given *ad-libitum*. Four chickens were sacrificed by cervical dislocation at day 0pi in group C followed by days 14 and 28pi in all groups. The body weight and blood were collected prior to sacrifice. On necropsy, the gross lesions were recorded and samples of trachea, liver and gizzard were collected and fixed in 10% buffered formalin for histological examination. The FAdV antibody titre was determined by enzyme linked immunoabsorbent assay (ELISA) test using commercial kit (Bio-Chek, UK, Ltd.) based on manufacture's recommendation. The animal study was conducted under approval of Institutional Animal Care and Use Committee (IACUC), Universiti Putra Malaysia with AUP No. R076/2015. DOI: dx.doi.org/10.17504/protocols.io.8ydhxs6.

**Table 2. Thirty avian adenovirus strains for fiber gene nucleotide and amino acids were used in the phylogenetic tree analysis.**

| No. | Strains | Accession number | | Group[a] | References |
|---|---|---|---|---|---|
| | | Nucleotide | Amino acid | | |
| 1 | CELO | U46933 | AAC54918 | A | S1 File [EE] |
| 2 | OTE | FN557186 | CBH20110 | A | S1 File [FF] |
| 3 | PL/060/08 | GU952108 | ADF57266 | A | S1 File [GG] |
| 4 | 08–3622 | FN557184 | CBH20108 | A | S1 File [HH] |
| 5 | 340 | FR872928 | CCB84856 | B | S1 File [II] |
| 6 | B1-7 | KU342001 | ANG08832 | C | S1 File [F] |
| 7 | Bareilly | FJ949088 | ACR54094 | C | S1 File [JJ] |
| 8 | Kr-Yeoju | HQ709232 | ADV35562 | C | S1 File [KK] |
| 9 | Kr-Gunwi | HQ709231 | ADV35561 | C | S1 File [LL] |
| 10 | AG234 | HE608153 | CCE39400 | C | S1 File [MM] |
| 11 | INT4 | FR872911 | CCB84839 | C | S1 File [NN] |
| 12 | ON1 | GU188428 | ADQ39072 | C | S1 File [OO] |
| 13 | C2B | HE608154 | CCE39405 | C | S1 File [PP] |
| 14 | SR48 | LN907576 | CUT98205 | D | S1 File [QQ] |
| 15 | 685 | KT862805 | ANJ02352 | D | S1 File [RR] |
| 16 | SR49 | LN907578 | CUT98207 | D | S1 File [SS] |
| 17 | A-2A | NC_000899 | NP_050293 | D | S1 File [M] |
| 18 | 380 | LN907574 | CUT98203 | D | S1 File [TT] |
| 19 | 05-50052-2924-1 | JQ034217 | AFD32284 | D | S1 File [UU] |
| 20 | 05-50052-3181 | JQ034218 | AFD32285 | D | S1 File [VV] |
| 21 | CR119 | LN907584 | CUT98213 | E | S1 File [WW] |
| 22 | YR36 | LN907582 | CUT98211 | E | S1 File [XX] |
| 23 | TR59 | KT037703 | ANQ43480 | E | S1 File [YY] |
| 24 | 764 | KT037711 | ANQ43488 | E | S1 File [ZZ] |
| 25 | HG | GU734104 | ADE58406 | E | S1 File [Y] |
| 26 | CFA3 | FAU40588 | AAC55302 | E | S1 File [AAA] |
| 27 | CFA40 | AF155911 | AAF17339 | E | S1 File [BBB] |
| 28 | UPM04217 | KU517714 | ANA50324 | E | S1 File [BB] |
| 29 | Duck adenovirus | KF286430 | AGS11279 | N/A | S1 File [CC] |
| 30 | Turkey adenovirus | AC_000016 | AP_000495 | N/A | S1 File [DD] |

N/A = not applicable

[a] FAdV comprised of 5 molecular groups designated as letter A, B, C, D and E

## Statistical analysis

Mean body weight and antibody titre were analyzed by analysis of variance (ANOVA) using SPSS version 22 with significant value $p<0.05$. The significant data was determined further by Tukey honest significant difference test. Independent T-test was performed to compare between two group means.

## Results

### Cytopathic effects (CPEs)

The uninfected primary CEL cells showed large irregular epithelial island of cells comprising of hepatocytes and surrounded by network of fibroblasts (Fig 1(A)). At first passage after inoculated with FAdV isolate, CPE was not observed until day 3 pi. From second to 35th passages,

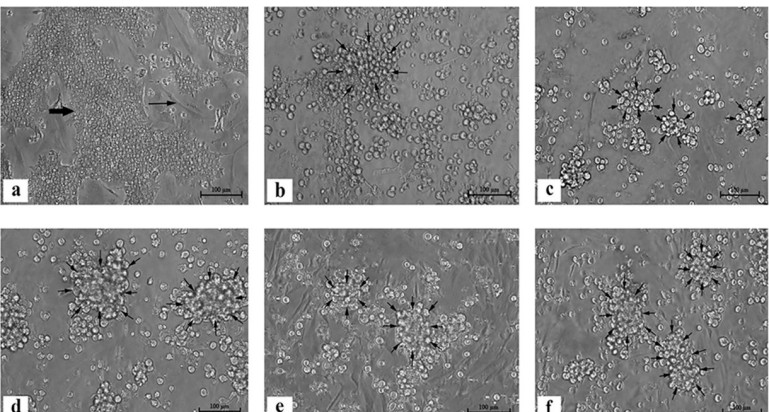

**Fig 1. Cytopathic effects of chicken embryo liver (CEL) cells following inoculation with FAdV isolate, UPM1137.**
(a) Uninfected primary CEL cells with large irregular epithelial islands (thick arrow) surrounded by network of fibroblast (line arrows) after 4 days culture. (b) Passage 2 (CEL2) infected cells with refractile, rounding and clumping of cells (arrows) at 26 hours pi. (c) Passage 10 (CEL10) infected cells with refractile, rounded and clumping of cells (arrows) at 32 hours pi. (d) Passage 20 (CEL20) infected cells with clumping of (arrows) at 52 hours pi. (e) Passage 30 (CEL30) infected cells with clumping of cells at 62 hours pi. (f) Passage 35 (CEL35) infected cells with refractile, rounding and clumping of cells on entire monolayer (arrows) at 70 hours pi, Scale bar = 100μm.

CPE was detected majority in epithelial islands in the form of refractile, rounding, clumping and detachment of cells from the monolayer of the flasks. Fibroblasts were largely unaffected and remained normal throughout the trial. Early CPE was recorded started from second to 19[th] passage onwards within 24 to 48 hours pi. The complete CPE was detected at earliest period at 26 hours pi from CEL2 to CEL8 (Fig 1(B)) followed by 32 hours pi at CEL9 to CEL13 (Fig 1(C)). From CEL14 to CEL19, complete CPE was recorded at 42 hours pi. However, CPE formation was delayed from CEL20 to CEL35 within 48 to 72 hours pi. Beginning at CEL20 to CEL22, CPE was completed at 52 hours pi (Fig 1(D)). The period gradually increased at CEL23 to CEL27 at 56 hours pi followed by CEL28 to CEL32 at 62 hours pi (Fig 1(E)). The slowest complete CPE was recorded from CEL33 to CEL35 at 70 hours pi (Fig 1(F)).

In early stage of infection, cells were rounded and refractile at 12 hours pi. At 12 hours later (24 hours pi), infected cells were aggregated and form clumping of cells. At late stage of infection, complete CPE (>80%) was recorded within 24 to 48 hours pi with detachment of cells from monolayer flasks. As compared to passage 20 (CEL20) onwards, infected cells begun to refractile and rounded at 24 hours pi and form clumping on 12 hours later (36 hours pi). Detachment of cells with complete CPE was observed within 48 to 72 hours pi. Non-infected cultures remained normal throughout study period.

### Haematoxylin and eosin staining

The uninfected cells were remained normal throughout the trial (Fig 2(A)). Numerous basophilic intranuclear inclusion bodies were observed in the infected cells at 48 hours pi (CEL35) (Fig 2(B)).

### Virus titration of propagated isolates

The virus titre was highest at passage 5 (CEL5) with $10^{7.6}$TCID$_{50}$/mL. For subsequence passages at CEL10 to CEL30, the titre ranging from $10^{6.2}$TCID$_{50}$/mL to $10^{6.8}$TCID$_{50}$/mL. The virus titre was $10^{6.7}$TCID$_{50}$/mL at passage 35 (CEL35).

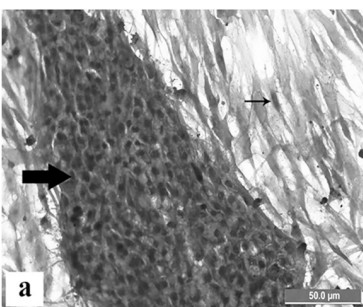
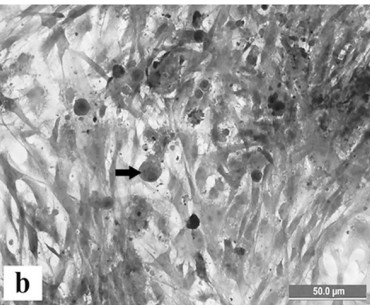

**Fig 2. Chicken embryo liver (CEL) cells stained with haematoxylin and eosin for detection of basophilic intranuclear inclusion bodies.** (a) Normal CEL cells comprising epithelial island of hepatocytes (thick arrow) surrounded by fibroblasts (thin arrow) at 48 hours post cultured. (b) Infected CEL cells by FAdV isolate (CEL35) at 48 hours pi with CPE of cells detachment and presence of basophilic intranuclear inclusion bodies (thick arrow) surrounded by unaffected fibroblast, Scale bar = 50.0μm.

## Molecular detection by PCR

The original isolate (E2) and CEL cells propagated isolates from the first passage (CEL1) to 35<sup>th</sup> passages (CEL35) were positive for FAdV with amplified PCR fragment size of 1219bp, 1319bp and 1124bp using H1/H2, H3/H4 and FibF/FibR primers, respectively (Fig 3(A), 3(B) and 3(C)).

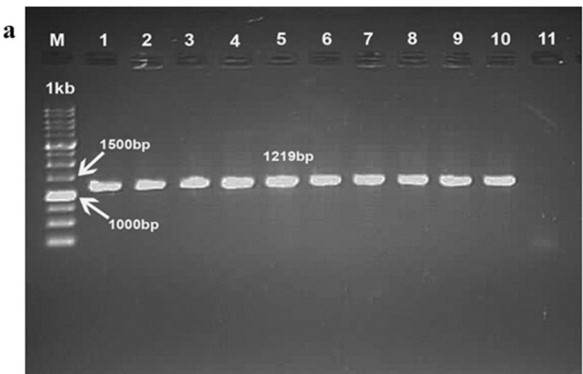
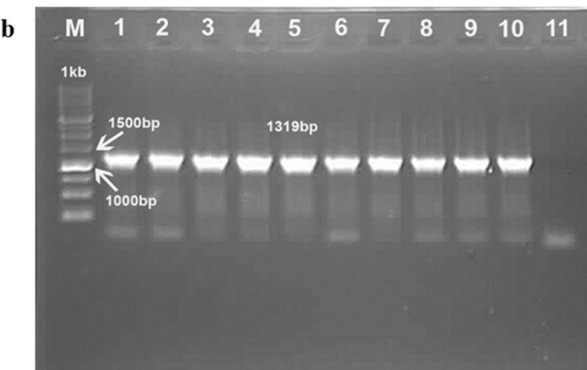
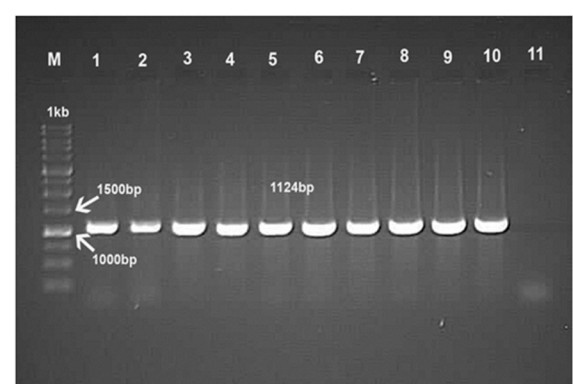

**Fig 3.  Electrophoresis of polymerase chain reaction products amplifying hexon and fiber gene regions of FAdV using primers, H1/H2, H3/H4 and FibF/FibR with fragment sizes of (a): 1219bp, (b): 1319bp, and (c): 1124bp in 1% agarose gel from sample of the original passage (E2) and passaged FAdV isolates in chicken embryo liver cells.** Lane M: 1kb DNA Marker (iNtRON), Lane 1: Positive control (UPM04217), Lane 2: Original isolate (E2), Lane 3: CEL1 FAdV, Lane 4: CEL5 FAdV, Lane 5: CEL10 FAdV, Lane 6: CEL15 FAdV, Lane 7: CEL20 FAdV, Lane 8: CEL25 FAdV, Lane 9: CEL30 FAdV, Lane 10: CEL35 FAdV, Lane 11: Negative control.

## Nucleotide, amino acids sequences and phylogenetic tree analysis

Analysis of partial hexon and fiber gene nucleotides in FAdV isolate from E2 or namely as UPM1137E2 was confirmed that the isolate belong to FAdV species E and showing 97% to 99% identity with the UPM04217 and 764 strains. The nucleotide of hexon and fiber genes with 1166bp and 1094bp length, respectively, in FAdV isolate throughout 35th consecutive passages at CEL5, CEL10, CEL15, CEL20, CEL25, CEL30 and CEL35 belong to similar species and showing 99% identity with UPM1137E2 following propagation in CEL cells. Several nucleotide (nt) and amino acid (aa) changes were detected in FAdV isolate following 35th consecutive passages in CEL cells. The hypervariable region in hexon gene, L1 loop, was determined to locate the changes specifically in every five consecutive passages and was identified at position 6 to 601 in nucleotides sequence and at position 2 to 199 in amino acid sequence.

There are 1nt base substitution at position 90(T-C) within L1 loop at CEL5 to CEL35 compared to E2 isolate before adaptation in CEL cells. Additional one nt change was detected in CEL5 (GenBank: **KY305943**), CEL10 (GenBank: **KY305944**) and CEL15 (GenBank: **KY305945**) at three different positions in each passage at 107(C-T), 1128(G-C) and 732(A-G), respectively. Those nt changes in CEL10 and CEL15 resulting alteration in deduced amino acid at position 376(A-P) and 244(M-V), respectively.

For subsequence passage, there are additional 2nt bases changes in CEL20 (GenBank: **KY305946**) at position 227(C-T) and 1080(A-G) resulting 1aa substitution at position 360 (I-V). Major hexon gene changes were detected at CEL25 (GenBank: **KY305947**) involving other 4nt bases at position 142(A-G), 292(C-T), 368(A-G) and 1122(A-C) along with 3aa in deduced amino acid at position 47(N-S), 97(P-L) and 374(T-P).

Propagation of UPM1137 isolate at passage 30 (CEL30) (GenBank: **KY305948**) in CEL cells resulting 2nt bases substitution at position 90(T-C) and 274(A-G) with changing in deduced amino acid at position 91(E-G). The highest passage, CEL35 (GenBank: **KY305949**) produced 4nt bases changes in L1 loop at position 90(T-C), 133(A-T), 400(C-T) and 556(T-A) which lead to substitution in 3aa at position 44(D-V), 133(S-F) and 185(V-E) after serial passages in CEL cells.

Analysis on fiber gene revealed several molecular changes involving tail, shaft and knob regions at position 1 to 66, 67 to 907 and 946 to 1094, respectively in nucleotide sequences. In deduced amino acid, those three regions were located at position 1 to 22, 23 to 303 and 316 to 364, respectively. Adaptation of UPM1137 at passage 5 (CEL5) (GenBank: **KY305951**) in CEL cells resulting four nt bases changes at position 31(T-C), 616(A-G), 652(G-A) and 963(A-G) with 3aa substitutions in deduced amino acid at position 11(S-P), 206(T-A) and 218(D-N). Following another five passaged in CEL cells, only 1nt base change was detected at position 971(A-T) resulting substitution of 1aa at location 324(Q-L) in CEL10 (GenBank: **KY305952**). The changes continually noticed in CEL15 (GenBank: **KY305953**) at 3 different position, 469 (C-T), 483(C-T) and 732(A-T) in nt sequence along with 1aa change at position 157(P-S).

Propagation of FAdV isolate at CEL20 (GenBank: **KY305954**) onwards induced 2nt bases substitutions at position 1050(A-C) and 1078(G-C) which resulting 1aa substitution at position 360(A-P). Another 2nt bases was substituted in CEL25 (GenBank: **KY305955**) at the region 839(A-G) and 1062(A-C) with 1aa change at position 280(N-S). Substitution of nt base at position 1062 were also consistent at CEL30 and CEL35. Additionally 2nt bases changes were detected in CEL30 (GenBank: **KY305956**) at position 295(A-G) and 680(T-C) which induced 2aa substitutions at position 99(T-A) and 227(L-S). Other 3nt bases was substituted at position 556(T-C), 821(T-A) and 1042(A-C) in CEL35 (GenBank: **KY305957**), which distinct from previous passage at CEL30. At deduced amino acid, 4aa changes were noticed at position 189(L-P) and 274(F-Y) in shaft region and in knob region at position 348(T-P) and 360(A-P).

Phylogenetic tree analyses of 38 aligned aa based on hexon and fiber gene revealed classification of FAdV into five major groups from A to E species based on current ICTV nomenclature (Figs 4 and 5). The original FAdV isolate (E2) and the propagated FAdV isolates from every fifth passages were derived from species E and shared common ancestor with other FAdV-8 strains such as UPM08136, UPM08158, UPM04217, 764, HG, 430–06 and Australian FAdV vaccine. On the other hand, both duck and turkey adenovirus were not related to FAdV species which remained out group from generated phylogram and assigned with other genus of avian adenovirus group.

## Pathogenicity and immunogenicity of attenuated FAdV, CEL35 isolate in SPF chickens

Neither the control nor the inoculated chickens with FAdV (CEL35) via oral or subcutaneous route exhibited any clinical signs, gross and histological lesions associated with FAdV infection throughout the trial (Fig 6(A), 6(B), 6(C) and 6(D)). The body weight of chickens in group A was significantly increased ($p < 0.05$) from 70 ± 3g to 391 ± 26g at day 0 to 28pi, respectively. A similar pattern of body weight increment was recorded in groups B and C. However, there were no significant difference ($p > 0.05$) on body weight in all groups of chickens throughout the trial. The FAdV antibody titre was not detected in group C throughout the trial. The antibody titre in group A was significantly ($p < 0.05$) increased to 316 ± 118 and 163 ± 17 at days 14 and 28pi, respectively when compared to the control group. The antibody titre in group B also increased to 648 ± 188 and 324 ± 85 at days 14 and 28 pi respectively. There were no significant different ($p > 0.05$) of antibody titre between route of inoculation either via oral or subcutaneous route.

## Discussion

The study demonstrated that FAdV isolate (UPM1137) was successfully adapted and propagated for 35[th] consecutive passages in primary CEL cells with typical CPEs of FAdV infection. The virus caused basophilic intranuclear inclusion bodies with refractile, rounding, clumping and detachment of CEL cells. However, the CPE was not observed until day 3 pi at the first passage. Early CPE was recorded started from second to 19[th] passages within 24 to 48 hours pi and a slight delayed within 48 to 72 hours pi from CEL20 to CEL35. The virus titre was highest at passage 5 ($10^{7.6}$TCID$_{50}$/mL) and remained high at the subsequence passages with the virus titre of $10^{6.7}$TCID$_{50}$/mL at passage 35. The development of CPE within 24 to 72 hours pi in CEL cells infected with FAdV reflects the sensitivity of this medium for FAdV replication even until passage 35 which is higher than previously reported [13, 14]. As compared to other cell lines such as Vero cells, its take more than 4 days pi for CPE to appear which is time consuming for propagation of the isolate [16]. These sensitive medium resembled in vivo conditions since nucleus of liver cells are the viral tropism for FAdV replication. Nevertheless, liver comprising high level expression of coxsackievirus-adenovirus receptor (CAR) and integrin receptor for adenovirus fiber binding to trigger infection [24, 25]. As a result, CPE formation with massive destruction of monolayer culture within early period of time was observed in this study as indicator of viral replication. With the ability of FAdV isolate to replicate and propagate at high passage level and produce high virus titre, it was demonstrated that CEL cells is a suitable medium for propagation and attenuation of the virus.

The present study revealed substitution of nucleotide base at position 90 from cytosine to thymine in L1 loop of hexon gene was detected in FAdV passaged isolate in CEL cells. It seems that these changes are crucial for the isolate to remain viable and growth in the CEL cells for 35 passages since the changes was not seen in original isolate (E2).

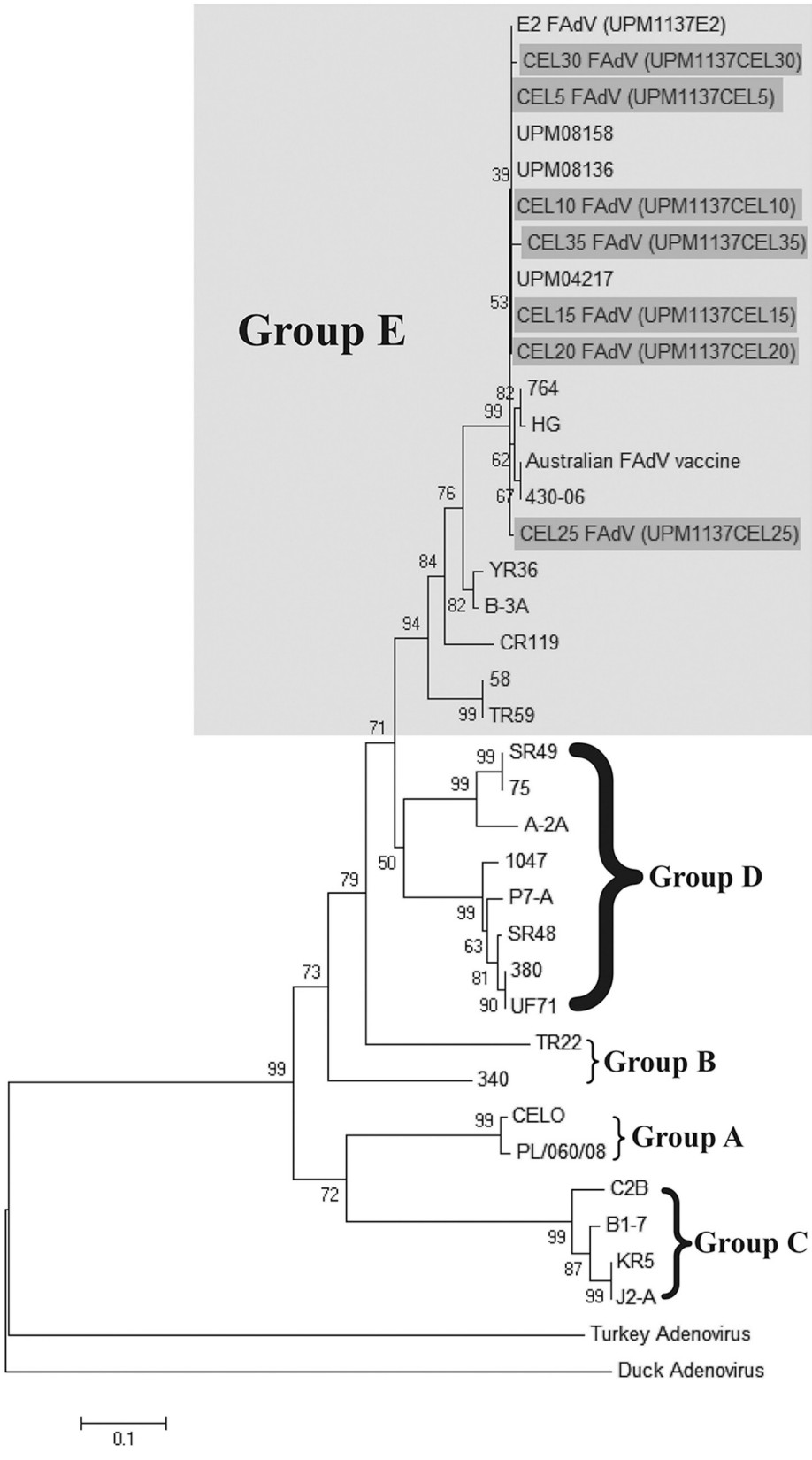

**Fig 4. Phylogenetic tree of 198 amino acid residues based on L1 loop hexon gene shown the passaged FAdV isolates in CEL cells (shaded in dark gray) were classified under group E species (shaded in light gray).**

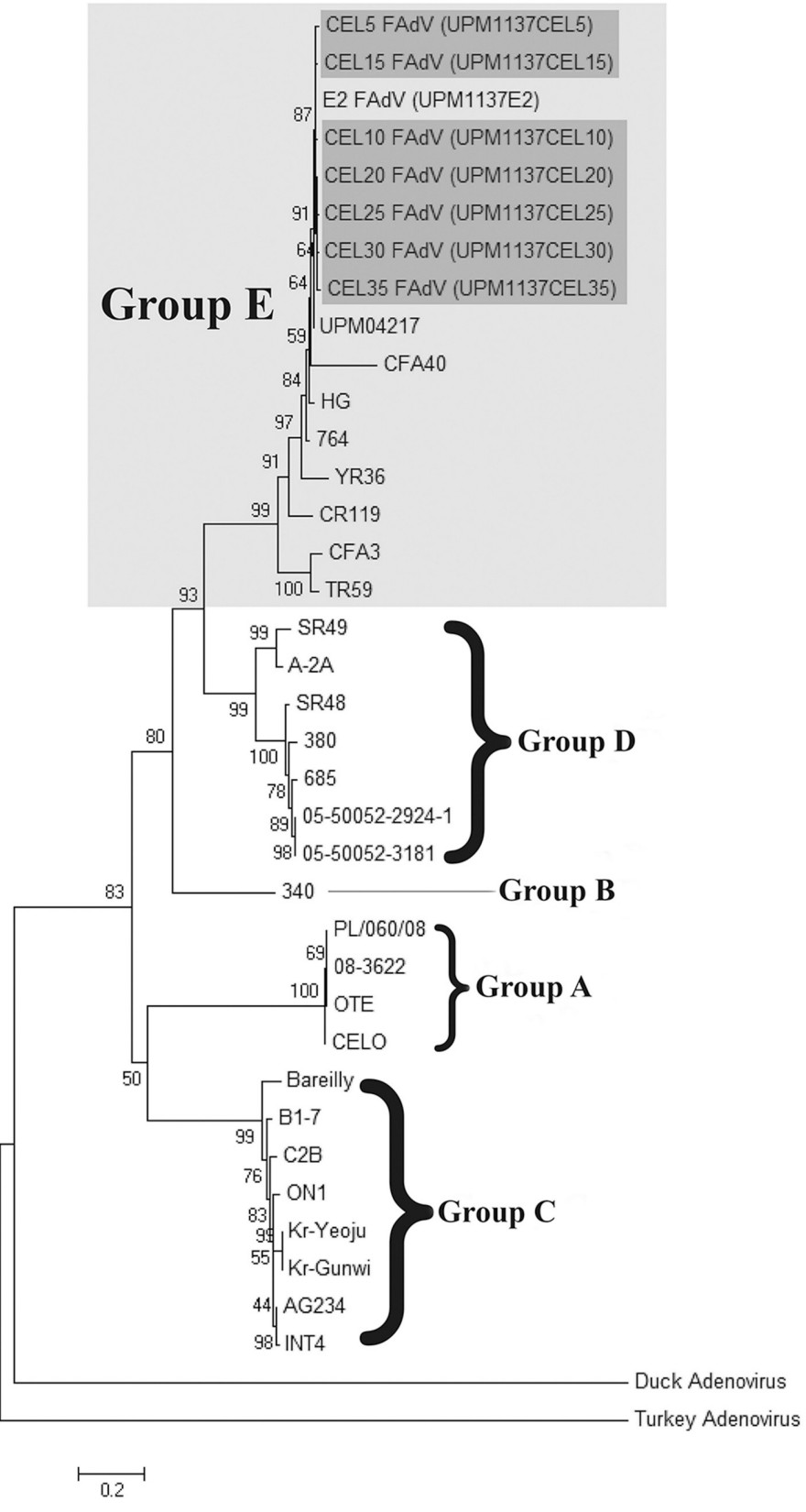

**Fig 5. Phylogenetic tree of amino acid residues based on fiber gene shown the passaged FAdV isolates in CEL cells (shaded in dark gray) was classified under group E species (shaded in light gray).**

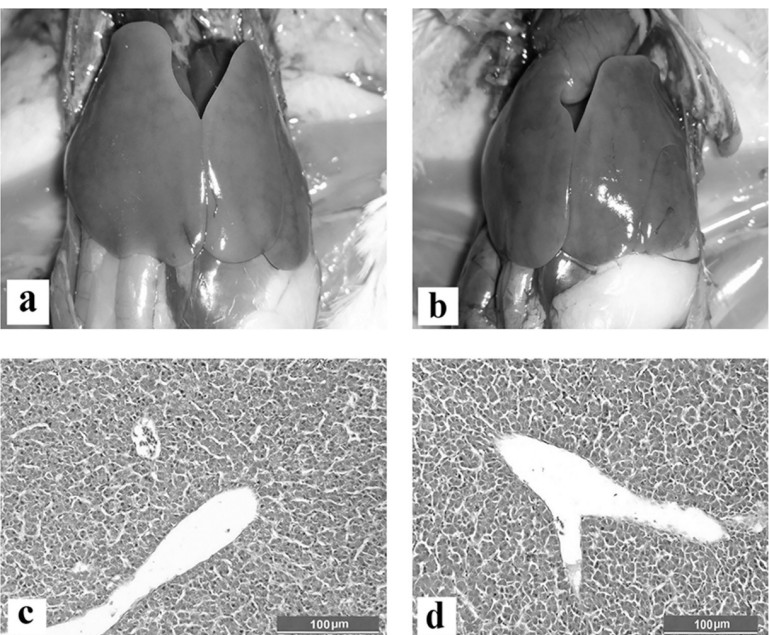

**Fig 6. SPF chicken at day 14pi showed normal gross and histological structure of liver in all group.** (a) Control group. (b) Group B of chicken inoculated with FAdV isolate subcutaneously. (c) Control group. (d) Group B of chicken inoculated with FAdV isolate subcutaneously. HE, Scale bar = 100μm.

Additionally, in fiber gene, substitution of amino acid at position 360 from alanine (A) to proline (P) resulted delaying in CPE formation within 48 to 72 hours pi in passaged isolate from CEL20 onwards. As compared to early 15 passages, CPE produced within 24 to 48 hours without interference at this amino acid region. The current finding was proved that serial passages of FAdV in CEL cells exhibit marker gene for adaptation and exposed molecular changes at the area encoded for virus infectivity.

Molecular changes were significant at highest passage (CEL35) in both L1 loop and knob region in hexon and fiber gene, respectively. Substitutions of deduced amino acid 44(D-V), 133(S-F) and 185(V-E) in L1 loop region along with amino acid 348(T-P) in knob of fiber gene in CEL35 are indicator of marker for the virus attenuation in CEL cells since these changes were not detected in other early passages.

Based on present findings, modification of knob region reduced binding affinity towards host cells receptor which affect virus infectivity in CEL and SPF chickens as compared to other part of fiber gene [10, 26, 27]. It shown that low amino acid identities were determined in knob region between high and low virulent FAdV isolates as reported in previous study and corroborated with current findings [17]. Additionally, mutation in L1 loop of hexon gene also resulting virus attenuation in embryonated chicken eggs [12]. Several changes occurs in L1 loop at CEL20 to CEL35 suggested that intervention at this region in hexon gene structure by serial passages in CEL cells might reduce virus capability to replicate in the cells with delayed in period of CPE formation.

Propagation of FAdV isolate at passages CEL5, CEL10 and CEL15, induce minimal number of nucleotide and amino acid changes in hexon and fiber genes at variable region. Interestingly, as increased passage level, there are increased in number of changes ranging from 2 to 5nt bases in hexon and 2 to 6nt bases in fiber gene from CEL20 onwards. It was demonstrated that molecular changes in all passaged isolates were not interfere antigenicity of FAdV. Based

on FAdV classification, the passaged isolate at highest passage, CEL35 remained in the group E species under serotype 8b in hexon and fiber gene.

It was confirmed that the CEL35 isolate was attenuated and non-pathogenic in SPF chickens. Neither mortality nor gross and histological lesions were observed in all chickens throughout the trial. A similar finding was observed in previous studies following inoculation of passaged isolate with high molecular changes [12, 28]. It was shown that substitution of amino acids in CEL35 isolate at position 44, 133 and 185 in L1 loop of hexon gene and at position 348 in knob of fiber gene affect virus infectivity in SPF chickens and was identified as marker gene for attenuation. The CEL35 isolate is immunogenic and able to induce antibody titre at days 14 and 28pi when the virus was inoculated either via subcutaneous or oral route in SPF chickens.

## Conclusion

It was concluded that consecutive passages of FAdV serotype 8b isolate in primary CEL cells can induce viral attenuation with molecular changes in major capsid proteins and affect virus infectivity in cell culture and SPF chickens.

## Supporting information

**S1 File. GenBank databases for Fowl Adenovirus sequence references.**
(DOCX)

## Acknowledgments

The authors would like to thanks to Ministry of Education, and Ministry of Science and Technology, Malaysia, for technical support.

## Author Contributions

**Conceptualization:** Mohd Hair Bejo, Abdul Rahman Omar, Aini Ideris, Nurulfiza Mat Isa.

**Formal analysis:** Norfitriah Mohamed Sohaimi.

**Funding acquisition:** Mohd Hair Bejo.

**Investigation:** Norfitriah Mohamed Sohaimi, Mohd Hair Bejo.

**Methodology:** Norfitriah Mohamed Sohaimi, Mohd Hair Bejo.

**Project administration:** Mohd Hair Bejo.

**Software:** Norfitriah Mohamed Sohaimi.

**Supervision:** Mohd Hair Bejo, Abdul Rahman Omar, Aini Ideris, Nurulfiza Mat Isa.

**Validation:** Mohd Hair Bejo, Abdul Rahman Omar, Aini Ideris, Nurulfiza Mat Isa.

**Visualization:** Norfitriah Mohamed Sohaimi, Mohd Hair Bejo, Abdul Rahman Omar, Aini Ideris, Nurulfiza Mat Isa.

**Writing – original draft:** Norfitriah Mohamed Sohaimi.

**Writing – review & editing:** Norfitriah Mohamed Sohaimi, Mohd Hair Bejo.

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
