## [Decision Letter · Decision Letter 0]

23 Sep 2019

PONE-D-19-20512

MOLECULAR CHARACTERIZATION OF FOWL ADENOVIRUS ISOLATE OF MALAYSIA ATTENUATED IN CHICKEN EMBRYO LIVER CELLS AND IT’S PATHOGENICITY AND IMMUNOGENICITY IN SPECIFIC PATHOGEN FREE CHICKENS

PLOS ONE

Dear Prof Bejo,

Thank you for submitting your manuscript to PLOS ONE. After careful consideration, we feel that it has merit but does not fully meet PLOS ONE’s publication criteria as it currently stands. Therefore, we invite you to submit a revised version of the manuscript that addresses the points raised during the review process.

I am delighted to accept your manuscript for publication in PLOS ONE with minor revisions, also, my apologies for the delay. It took much longer than I anticipated to recruit reviewers for your manuscript. I agree with the reviewer recommendation to shorten the title, but the decision is solely yours and will not effect the decision to publish your findings. 

We would appreciate receiving your revised manuscript by Nov 07 2019 11:59PM. To enhance the reproducibility of your results, we recommend that if applicable you deposit your laboratory protocols in protocols.io, where a protocol can be assigned its own identifier (DOI) such that it can be cited independently in the future. For instructions see: http://journals.plos.org/plosone/s/submission-guidelines#loc-laboratory-protocols

We look forward to receiving your revised manuscript.

Kind regards,

Negin P. Martin, Ph.D.

Academic Editor

PLOS ONE

Reviewers' comments:

Reviewer's Responses to Questions

**Comments to the Author**

1. Is the manuscript technically sound, and do the data support the conclusions?

Reviewer #1: Yes

Reviewer #2: Yes

2. Has the statistical analysis been performed appropriately and rigorously? 

Reviewer #1: Yes

Reviewer #2: Yes

3. Have the authors made all data underlying the findings in their manuscript fully available?

Reviewer #1: Yes

Reviewer #2: Yes

4. Is the manuscript presented in an intelligible fashion and written in standard English?

Reviewer #1: Yes

Reviewer #2: Yes

5. Review Comments to the Author

Reviewer #1: Fowl adenovirus (FAdV) is the causative agent of inclusion body hepatitis (IBH) in chickens with significant economic losses. Authors conducted the study to attenuate and determine the molecular characteristic of FAdV isolate (UPM1137) of Malaysia passages in primary chicken embryo liver (CEL) cells. The cytopathic effect (CPE) was recorded and the present of the virus was detected by polymerase chain reaction (PCR). Authors also have detected intranuclear inclusion bodies in primary CEL cells by hematoxylin and eosin staining. The next step of conducted study the authors evaluated virus titration and DNA extraction, PCR and cloning. Nucleotide and amino acid changes were determined and a phylogenetic tree was constructed.

The pathogenicity and immunogenicity of the virus at passage 35 (CEL35) with virus titre of 10 6.7TCID50/mL was determined in day old specific pathogen free (SPF) chicks via oral or subcutaneous route of inoculation.The study demonstrated that the FAdV isolate was successfully propagated and attenuated in CEL cells up to 35th consecutive passages (CEL35) with delayed of CPE formation within 48 to 72 post inoculation (pi) from CEL20 onwards. The virus caused typical CPE with basophilic intranuclear inclusion bodies, refractile and clumping of cells.

Statistical analysis have been conducted and independent T-test was performed. The virus was classify to serotype 8b with substitution of amino acid at position 44, 133 and 185 in L1 loop of hexon gene and in knob of fiber gene at position 348 and 360 at CEL35. It is non-pathogenic, but immunogenic in SPF chickens. It was concluded that the FAdV isolate was

successfully attenuated in CEL cells with molecular changes in major capsid proteins which affect it’s infectivity in cell culture and SPF chickens.

The manuscript is well written and has an impact in front of adenovirus molecular investigation. I am for the publication this manuscript in PLOS one Journal.

Reviewer #2: 1- The manuscript title "MOLECULAR CHARACTERIZATION OF FOWL ADENOVIRUS ISOLATE OF MALAYSIA ATTENUATED IN CHICKEN EMBRYO LIVER CELLS AND IT’S PATHOGENICITY AND IMMUNOGENICITY IN SPECIFIC PATHOGEN FREE CHICKENS" is too long it may be changed and shortened?

see one example below.

"MOLECULAR CHARACTERIZATION OF A MALAYSIAN FOWL ADENOVIRUS ISOLATE ATTENUATED IN CHICKEN EMBRYO LIVER CELLS"

2- You should also describe about SPF chickens, either they were SPF broilers or layers?

6. PLOS authors have the option to publish the peer review history of their article (what does this mean?). If published, this will include your full peer review and any attached files.

Reviewer #1: Yes: Niczyporuk Jowita Samanta

Reviewer #2: Yes: Dr. M. Salah-ud-Din Shah

---

## [Author Response · Author response to Decision Letter 0]

9 Nov 2019

1) The manuscript title "MOLECULAR CHARACTERIZATION OF FOWL ADENOVIRUS ISOLATE OF MALAYSIA ATTENUATED IN CHICKEN EMBRYO LIVER CELLS AND IT’S PATHOGENICITY AND IMMUNOGENICITY IN SPECIFIC PATHOGEN FREE CHICKENS" is too long it may be changed and shortened?

see one example below.

"MOLECULAR CHARACTERIZATION OF A MALAYSIAN FOWL ADENOVIRUS ISOLATE ATTENUATED IN CHICKEN EMBRYO LIVER CELLS"

Answer: The study involved with attenuation of Fowl adenovirus in chicken embryo liver cells (CEL) and characterized molecularly the virus characteristic resulted from attenuation of virus in CEL. In addition to molecular characterization, the study also determine the pathogenicity and immunogenicity of the virus in chickens. Thus, the original title reflect more clearly on the study and contents of the manuscript. It is advisable to remove the specific pathogen free (SPF) to shorten the title. The CEL was prepared form liver of SPF chicken embryo as well as the pathogenicity and immunogenicity were conducted in SPF chickens. SPF was not mentioned in CEL. Thus, it justify to remove SPF for the chickens.

New title: "MOLECULAR CHARACTERIZATION OF FOWL ADENOVIRUS ISOLATE OF MALAYSIA ATTENUATED IN CHICKEN EMBRYO LIVER CELLS AND IT’S PATHOGENICITY AND IMMUNOGENICITY IN CHICKENS"

2) You should also describe about SPF chickens, either they were SPF broilers or layers?

Answer: The SPF chickens used in the study was from SPF White Leghorns layer chickens. It is included in the revised manuscript.

---

## [Editor Report · Decision Letter 1]

14 Nov 2019

MOLECULAR CHARACTERIZATION OF FOWL ADENOVIRUS ISOLATE OF MALAYSIA ATTENUATED IN CHICKEN EMBRYO LIVER CELLS AND IT`S PATHOGENICITY AND IMMUNOGENICITY IN CHICKENS

PONE-D-19-20512R1

Dear Dr. Bejo,

We are pleased to inform you that your manuscript has been judged scientifically suitable for publication and will be formally accepted for publication once it complies with all outstanding technical requirements.

With kind regards,

Negin P. Martin, Ph.D.

Academic Editor

PLOS ONE
---

## [Editor Report · Acceptance letter]

19 Dec 2019

PONE-D-19-20512R1 

Molecular Characterization of Fowl Adenovirus Isolate of Malaysia Attenuated in Chicken Embryo Liver Cells and It’s Pathogenicity and Immunogenicity in Chickens

Dear Dr. Bejo:

I am pleased to inform you that your manuscript has been deemed suitable for publication in PLOS ONE. Congratulations! Your manuscript is now with our production department. 

With kind regards,

on behalf of

Dr. Negin P. Martin 

Academic Editor

PLOS ONE